# Redox Regulation of Salt Tolerance in *Eutrema salsugineum* by Proteomics

**DOI:** 10.3390/ijms241914518

**Published:** 2023-09-25

**Authors:** Jiawen Li, Xiaomin Yang, Fuqing Liu, Xinxin Liu, Tong Zhao, Xiufeng Yan, Qiuying Pang

**Affiliations:** 1Key Laboratory of Saline-Alkali Vegetation Ecology Restoration, Ministry of Education, College of Life Sciences, Northeast Forestry University, Harbin 150040, China; ljwcell@163.com (J.L.); yangxiaomin@nefu.edu.cn (X.Y.); fuqingliu@nefu.edu.cn (F.L.); xinxinnefu@163.com (X.L.); ztscience@126.com (T.Z.); 2Zhejiang Provincial Key Laboratory for Water Environment and Marine Biological Resources Protection, College of Life and Environmental Science, Wenzhou University, Wenzhou 325035, China

**Keywords:** *Eutrema salsugineum*, NaCl, TMT, redox proteomics, glutathione peroxidase

## Abstract

Salt stress severely restricts plant growth and crop production, which is accompanied by accumulation of reactive oxygen species (ROS) that disturb cell redox homeostasis and oxidize redox-sensitive proteins. *Eutrema salsugineum*, a halophytic species closely related to *Arabidopsis*, shows a high level of tolerance to salinity and is increasingly used as a model plant in abiotic stress biology. To understand redox modifications and signaling pathways under salt stress, we used tandem mass tag (TMT)-based proteomics to quantify the salt-induced changes in protein redox modifications in *E. salsugineum*. Salt stress led to increased oxidative modification levels of 159 cysteine sites in 107 proteins, which play roles in carbohydrate and energy metabolism, transport, ROS homeostasis, cellular structure modulation, and folding and assembly. These lists of unknown redox reactive proteins in salt mustard lay the foundation for future research to understand the molecular mechanism of plant salt response. However, glutathione peroxidase (GPX) is one of the most important antioxidant enzymes in plants. Our research indicates that *EsGPX* may be involved in regulating ROS levels and that plants with overexpressed *EsGPX* have much improved salt tolerance.

## 1. Introduction

Salt stress affects more than 800 million hectares of land (~6% of the world’s total arable area) [1], inhibiting various physiological processes such as photosynthesis, respiration, energy metabolism, and plant growth [2]. One major mechanism by which salt has an adverse impact on plant development and growth is that salt introduces excessive production of reactive oxygen species (ROS) and oxidative stress, which are associated with lipid peroxidation, dysregulation of the cell membrane and various cellular components in plants [3]. Understanding plant responses to salt stress at the molecular level is crucial to develop plants with sustained productivity at higher salt concentrations. However, a major knowledge gap in plant stress biology is how redox regulation contributes to the various adverse outcomes of abiotic stress including salt.

Recent advances in redox biology have led to the recognition of ROS as signaling molecules under normal physiological processes [4]. This paradigm shift of ROS from purely unwanted byproducts that cause cellular damage to key signaling players was accompanied by the discovery that ROS can modify redox-sensitive proteins at the post-translational modification level [5]. Redox-sensitive proteins act as molecular switches in controlling protein activity, stability, interactions, and cellular location [6]. Thus, changes in ROS production can result in changes in the oxidation levels of key redox-sensitive proteins, which in turn impact their cellular functions and the overall cellular responses [5]. Despite many previous studies having shown that salt stress can induce an increase in ROS plants, the extent of ROS-induced regulation at the protein modification level is largely unknown.

On the other end of redox regulation is the antioxidant system including key enzymes such as glutathione peroxidases (GPXs). GPXs maintain a low level of endogenous hydroperoxide (H_2_O_2_) using glutathione (GSH). GPXs have been extensively studied in multiple plant species including tobacco [7], *Arabidopsis thaliana*, *Oryza sativa*, and *Solanum lycopersicum* L. [8]. At the transcript level, the expression of GPXs was impacted by salt stress [9]. Other studies suggested a critical role of GPXs in ROS elimination in mitochondria and endosomes, key cellular compartments for ROS production [10].

*Eutrema salsugineum*, formerly known as *Thellungiella salsuginea* or *Thellungiella halophila*, is a halophyte of the Brassicaceae family. *E. salsugineum* can grow on saline soil in farmland and is widely cultivated in central Asia, North America, and China [11]. Its strong salt tolerance, along with other desirable features such as a small genome, short size, short life cycle, and self-pollination [12], makes it a rare halophyte model plant to study salt tolerance [13]. For example, transcriptomics and metabolomics revealed gene regulatory networks and metabolic pathways under salt stress in *E. salsugineum*, highlighting the importance of cellular redox homeostasis [14,15]. However, redox modifications on the *E. salsugineum* proteome have not been reported.

Here, we first quantitatively profiled the redox proteome of *E. salsugineum* under salt stress using tandem mass tags (TMT). The workflow also included a module to quantify the protein abundance, so that quantification information on both protein abundance and redox modifications can be integrated to identify the salt-induced changes. We found that differentially expressed proteins are involved in photosynthesis, metabolism, cell structure, protein synthesis, energy balance, stress, and defense. More importantly, a list of redox-responsive proteins were identified, highlighting processes that play important roles under salt treatment. One significant regulatory protein from the large-scale analyses was GPX. Thus, we experimentally evaluated the role of GPX from *E. salsugineum* in regulating stress tolerance.

## 2. Results

### 2.1. The Impact of Salt on the ROS Scavenging System

We first investigated the impact of NaCl (three doses of 150, 300, and 500 mM and three time points of 6 h, 24 h, and 7 d, respectively) on the ROS scavenging system in *E. salsugineum*. We found that treatment with 300 mM NaCl resulted in remarkable increases in O_2_^−^ and H_2_O_2_ (Figure 1A,B) and SOD activity (Figure 1D). A rapid rise in MDA production was also observed under 500 mM NaCl for 24 h and all conditions for 7 d regardless of the concentration. Quantification of the antioxidant enzyme system, including POD, CAT, and four key enzymes in the ascorbate–glutathione (AsA−GSH) cycle revealed a significant change in the activity of POD and CAT under 500 mM NaCl treatment (Figure 1E,F). Interestingly, the activities of enzymes in the AsA-GSH cycle varied depending on the treatment condition (Figure 1G,H).

Besides the antioxidant enzyme system, we also quantified nonenzymatic antioxidants including AsA, DHA, and GSH. We found an increase in AsA and a decrease in GSH under 300 mM NaCl for 24 h. Surprisingly, the level of GSSG, the oxidized form of GSH, was unchanged at all the time points (Figure 1H,I). These suggested that salt treatment can lead to a decrease in the GSH/GSSG ratio, which is crucial for cellular redox homeostasis. Because high levels and long durations of salt stress can cause severe oxidative damage to *E. salsugineum* plants, one needs to use a dosage that triggers the redox state but is not toxic to the system for the purpose of identifying salt-induced oxidative modifications. Based on our physiological data, we chose 300 mM NaCl at 24 h for redox proteomics analysis due to the accumulation of ROS and its non-toxicity.

### 2.2. Redox Proteomics Reveals Proteins Sensitive to Salt Stress in E. salsugineum Leaves

To quantify the cysteine proteome, we used an established method to measure both the protein abundance and the level of cysteine oxidation [16]. We identified 3735 unique cysteine-containing peptides in 1875 proteins. The mass errors of cysteine-containing peptides were less than 10 ppm, indicating the high accuracy of our MS data (Appendix A). Most peptides varied from 7–24 amino acids in length (Appendix A). A representative MS/MS spectra showed the presence of both TMT reagent reporter ions and b/y ion series (Appendix A).

We next performed differential analysis between the control and NaCl-treated groups (protein abundance change threshold of 1.2 times and *p*-value of 0.05 by *t*-test). Among the 752 quantifiable proteins (Appendix A), 68 were up-regulated and 39 were down-regulated in protein oxidation (Figure 2A, Appendix A). Examination of the redox proteomics dataset showed a total of 45 proteins whose abundances were also quantified (Figure 2B). Thus, we focused on these 45 proteins for further analysis.

The sequences of the 45 redox-regulated proteins were analyzed for disulfide bond formation, revealing four proteins with the potential to form intra-molecular disulfide bonds (Appendix A). The 45 significant redox-regulated proteins can be found in several cellular compartments: chloroplast (*n* = 13), cytoplasm (*n* = 13), extracellular matrix (*n* = 9), nucleus (*n* = 3), mitochondria (*n* = 3), vacuolar membrane (*n* = 1), and cytoskeleton (*n* = 1). This diversity in the subcellular localization suggested that these proteins may have key roles in a variety of cellular processes (Figure 3A). Indeed, GO enrichment analysis showed that the redox-regulated proteins were involved in response to cell, intracellular, cytoplasm, organic substance metabolic processes, cellular metabolic processes, and responses to stress (Figure 3B).

Hierarchical clustering revealed two main clusters: cluster I of 26 proteins mainly involved in transport, carbohydrate and energy metabolism, protein turnover, and chaperones, and cluster II of 19 proteins mainly involved in cell structure modulation, RNA processing and modification, coenzyme and inorganic ion transport and metabolism. Cluster I and II proteins showed an increase and decrease in protein oxidation, respectively, upon NaCl treatment. One protein of particular interest was glutathione peroxidase (GPX), which showed a significant oxidation under NaCl treatment (1.5 fold increase, *p*-value = 0.0061822). The protein abundance of GPX shows an increase under NaCl treatment. Due to its importance as a key antioxidant enzyme, GPX was selected for further functional study.

### 2.3. Ectopic Expression of EsGPX in Arabidopsis Enhances Salt Tolerance

To investigate the physiological role of *EsGPX* under abiotic stress, stable transgenic *Arabidopsis* overexpressing *EsGPX* was generated and homozygous T3 lines were obtained (Figure 4A). One representative line, designated as OE3, was used in the following experiments. First, we studied the salt-responsiveness in OE3, wild type (WT) plant, and *gpx*, a knockout mutant of GPX in *Arabidopsis*. Salt treatment for a week resulted in leaf yellowing and wilting in the *gpx* mutant (Figure 4B). By contrast, OE3 plants were greener and showed higher viability under the same condition. For seed germination, there was no discernible difference among the three lines under control conditions (Figure 4C,D). However, OE3 showed a higher germinated rate (38.93%) compared to WT (37.93%) and *gpx* (8%) under 200 mM NaCl (Figure 4C,D). We also tested the impact of salt on root elongation, in which 150 mM NaCl greatly hindered the root growth for all three lines (Figure 4E,F). However, the extent of inhibition was smaller in OE3 compared to the other two lines, indicating a positive role of GPX on salt response.

The overall fitness of the plant can be reflected by the chlorophyll content, in which we found no significant difference in the photosynthetic pigments (chlorophylls a/b and carotenoids) among the three lines under control conditions (Figure 5A–D). Under salt stress, however, all types of chlorophylls were dramatically higher compared to that in the WT and *gpx* plants. These findings suggest that GPX may alleviate salt stress-induced decrease in chlorophyll production.

### 2.4. Overexpressing EsGPX Alleviates Salt-Induced Oxidative Stress

To directly evaluate the impact of GPX on the ROS levels, we quantified O_2_^−^ and H_2_O_2_ in the leaf. Considerably lower levels of O_2_^−^ and H_2_O_2_ were found in OE3, as visualized by less blue staining with NBT and a lighter brown hue following DAB staining, respectively (Figure 5E). We then measured endogenous leaf H_2_O_2_ levels, which showed an increase in H_2_O_2_ following salt stress (Figure 5F). Significantly, the level of H_2_O_2_ in OE3 was much lower compared to the other two lines. Consistent with the phenotypical data, MDA levels in leaves of all strains were similar under standard growth conditions. However, a much lower level of MDA was observed in OE3 compared to that of the WT and *gpx* under salt challenge (Figure 5G). These data suggested that GPX can alleviate salt-induced oxidative stress.

The redox status of any organism is controlled not only by the ROS, but also by the antioxidant system. Thus, we next determined the activity of several enzymes including catalase, ascorbate peroxidase, and GSTs and the level of GSH (Figure 6). We found that POX was more active in *gpx* compared to WT under both the control and NaCl treatment conditions (Figure 6A). For APX, salt treatment led to an increase in WT and *gpx*, but a decrease in OE3 (Figure 6B). The SOD activity was lower in OE3 compared to WT under both conditions (Figure 6C). The most notable enzyme was POD, which showed comparable levels in the three lines under normal conditions, but a significantly higher activity in OE3 compared to the other two under salt stress (Figure 6D). The activity of CAT was slightly lower in OE3 compared to the WT under control conditions, but no significant difference was found under the salt treatment (Figure 6E). For GST, the three lines did not show any statistical differences under either condition (Figure 6F). By contrast, GR showed the most significant variations among the three lines in the order of WT > *gpx* > OE in both conditions (Figure 6G). Lastly, OE3 also exhibited higher levels of GSH under both the control and salt treatment.

## 3. Discussion

Previous studies have elucidated the regulatory mechanisms involved in the salt stress of *E. salsugineum* at the transcription level [17]. However, changes in gene expression may or may not lead to changes in protein abundance as other regulatory processes including post-translational modifications (PTMs) play an important role in controlling protein abundance [18,19,20]. One of the most important PTMs is redox regulation under oxidative stress. Despite the well-recognized role of ROS in plant stress response [21], the impact of ROS production on the redox proteome of *E. salsugineum* remains unclear. To tackle the two key knowledge gaps, we used proteomics and redox proteomics to delineate the proteins and pathways responsible for salt stress tolerance.

We found higher H_2_O_2_ levels under NaCl treatment (Figure 1), which was concordant with the activation of antioxidant-related enzymes including SOD (EUTSA_v10008927m) and MDHAR (Monodehydroascorbate reductase, EUTSA_v10023425mg). MDHAR catalyzes the reduction of monodehydroascorbic acid (MDHA) to ascorbic acid for removing excess oxygen free radicals. Upregulation of these antioxidant enzymes indicated their roles in the maintenance of ROS homeostasis in response to salt stress. We also identified redox-sensitive antioxidant proteins including peroxidase (EUTSA_v10004581mg; EUTSA_v10021123mg), glutathione s-transferase (GST, EUTSA_v10027346mg), glutathione peroxidase (GPX, EUTSA_v10017319mg), and succinic semialdehyde dehydrogenase (SSADH, EUTSA_v10018394mg). The diverse roles of these proteins indicated that multiple redox pathways were modulated under salt stress. For example, GPX can reduce H_2_O_2_ to H_2_O by consuming GSH [21], and GST can catalyze the reduction of organic hydroperoxides using GSH as the substrate. In addition, SSADH is involved in the γ-aminobutyrate shunt metabolic pathway and detoxifies ROS intermediates [22,23].

Our redox proteomics data revealed that several cell-wall-related enzymes were sensitive to salt treatment. For example, COBRA-like protein 5 (EUTSA_v10004931mg), chitinase 10-like (CTL, EUTSA_v10008465mg), and pectinesterase (EUTSA_v10020354mg) were more reduced after salt stress, while pectin acetylesterase (PAE, EUTSA_v10024796mg) was more oxidized upon salt treatment. These proteins participate in different aspects of cell structural modifications. For instance, COBRA regulates the direction of cell expansion [24]. CTL1 can be preferentially found in the cell wall of root tips and promotes root growth, radial swelling, lateral root proliferation and root hair density [25]. Pectinesterase is a ubiquitous cell wall pectin methylesterase that regulates post-germinative growth of seedlings by mobilizing seed storage proteins. PAEs control plant cell wall phenotypes by regulating the acetylation state of pectin polymers [26]. Collectively, our data indicated that cell wall dynamics can be controlled via redox-based mechanisms.

Other interesting proteins responsive to NaCl-induced oxidative stress were involved in protein synthesis, folding, and assembly. These include heat shock protein 90 (HSP 90; EUTSA_v10003676mg), 60S ribosomal protein L8 (RSL8; EUTSA_v10026029mg), 60S ribosomal protein L18a (RSL18; EUTSA_v10017304mg), 40S ribosomal protein S29-like (EUTSA_v10026741mg) and 40S ribosomal protein S5-1-like (EUTSA_v10017239mg) (Appendix A). Many of them are molecular chaperones, which facilitate the folding of newly translated proteins and the degradation of misfolded proteins [27,28].

NaCl stress may also cause redox changes in energy metabolism, and photosynthesis. This was evidenced by the changes in the oxidation state of glucanendo-1,3-beta-Glucosidase 9 (EUTSA_v10012621mg), O-Glycosyl hydrolases family 17 protein (EUTSA_v10010363mg), beta-galactosidase (EUTSA_v10024387mg), and glyceraldehyde-3-phosphate dehydrogenase (EUTSA_v10008131mg and EUTSA_v10004217mg). Another interesting protein was ferridodoxin (EUTSA_v10023742mg), which showed a higher oxidation level after NaCl treatment. Ferridodoxin is a major electron donor for the double bond reduction during the synthesis of the open-chain tetrapyrrole chromophore groups of photopigments [29]. This observation is consistent with the fact that photosynthesis is tightly regulated by redox control.

The omics data led us to focus on the role of *EsGPX* in salt response. Indeed, our results showed that overexpression of *EsGPX* protected plants from NaCl-induced oxidative stress by maintaining a high seed germination rate in the presence of 150 or 200 mM NaCl. Interestingly, OE-*EsGPX* seedlings showed a similar germination rate, root length, and chlorophyll contents compared to that of the WT, indicating that the protective role of *EsGPX* was activated under stress conditions (Figure 4E). The importance of GPX in salt tolerance was also confirmed by the hypersensitivity to salt in the *gpx* knockdown (Figure 5F). Moreover, our data were in accordance with the observation that GPXL is induced under salt and osmotic stress in *E. salsugineum* [30].

Our detailed biochemical analyses further revealed the key mechanism by which *EsGPX* protects plants from salt-induced oxidative stress. Overexpression of *EsGPX* increased the amount of total GSH in the transgenic plants, which showed a more negative redox potential than that of WT under the control condition. The difference was enlarged under salt stress, giving the transgenic plants a much stronger redox buffering capacity toward oxidative stress (Figure 6). Elevated GSH in the OE-*EsGPX* plants was also correlated with lower activity of antioxidant enzymes including CAT and GR, because these enzymes are usually active under oxidative stress. GSH is a main low molecular weight antioxidant. It participates in the Asc-GSH cycle to donate electrons to diverse antioxidant enzymes and can also modulate developmental responses via hormonal control [31]. The level of GSH determines the GSH/GSSG ratio, a key indicator of redox homeostasis [32]. It has been proposed that plant GPXs are critical redox sensors. However, it remains unclear how overexpression of GPX contribute to an enhanced GSH and altered redox status. It warrants further studies on the impact on the biosynthesis and consumption of GSH.

In summary, our data demonstrate that *EsGPX* enhances salt stress tolerance. One likely mechanism is to fine-tune the levels of ROS and GSH. This supports its role as a redox-sensing and ROS-scavenging protein in plants [33,34,35]. However, it remains unclear how overexpression of GPX contributes to an enhanced GSH and altered redox status. It warrants further studies on the impact on the biosynthesis and consumption of GSH.

## 4. Materials and Methods

### 4.1. Plant Materials and Treatments

*E. salsugineum* was a gift from Shandong Normal University. Seeds were rinsed with 75% ethanol for 10 min and 100% ethanol for 1–2 times. Seeds were kept on 1/2 MS medium at 4 °C in the dark for seven days before cultivation in a growth chamber. Once two young leaves emerged, the seedlings were transplanted to a potting mix (soil/vermiculite of 3/1) and cultured under a 16/8 h photoperiod. After 4 weeks, they were treated with NaCl of indicated concentrations for 24 h. Samples were frozen with liquid nitrogen and stored at −80 °C if not used immediately.

*Arabidopsis thaliana* ecotype Columbia (Col-0) was used as the wild type and for generating transgenic lines overexpressing GPX. The T-DNA insertion knockout mutant, *gpx* (SALK_082445C; Col-0 background), was obtained from the AraShare (https://www.arashare.cn/index/ (accessed on 1 July 2023)).

### 4.2. Vector Constructions and Plant Transformation

To over express *EsGPX*, the cDNA with full-length coding sequence was inserted into the pDONR222 vector and subsequently transferred into the pGWB405 vector (Thermo Fischer Scientific, Vilnius, Lithuania) [36]. Primers were listed in Appendix A. The Col-0 wild-type *Arabidopsis* plants were transformed using the infiltration method [37]. Homozygous overexpression lines were obtained by Kanosamine antibiotic selection of self-pollinated T1 and T2 plants. The level of *EsGPX* expression was checked in 10 transgenic lines by quantitative real-time polymerase chain reaction (RT-qPCR). Lines with the highest expression levels (Figure 4A) were chosen for further propagation and genetic analysis. T3 plants were used for further experiments.

### 4.3. Phenotypical Analysis of Seed Germination, Root Length, and Flowering

Seeds from wild-type (WT), *gpx*, and T3 transgenic plants were placed on 1/2 MS agar plates supplemented with 150 mM NaCl. Distilled water was used for mock treatment. The seed germination rate was determined at indicated time points. At the same time, the root length was observed with vertical culture for 7 days.

### 4.4. Sample Preparation of Proteomics

Samples were ground in liquid nitrogen into powder and then transferred to a 5 mL centrifuge tube. Four volumes of phenol extraction buffer (25 mM IMA, 1% Protease Inhibitor Cocktail) were added to the powder, followed by sonication three times on ice. Cellular debris was removed by centrifugation at 12,000× *g* at 4 °C for 10 min. The supernatant was collected and the protein concentration was determined using BCA (Beyotime Biotechnology, Shanghai, China).

For global proteomics, an aliquot of protein samples was reduced with 5 mM dithiothreitol for 30 min at 56 °C, then alkylated with 11 mM iodoacetamide for 15 min at room temperature in darkness. Samples were diluted with 100 mM TEAB until the urea concentration was less than 2 M. Trypsin was added at 1:50 trypsin-to-protein mass ratio. The first digestion was performed for overnight, followed by adding a 1:100 trypsin-to-protein mass ratio for a second 4 h digestion. The resulting peptides were desalted using Strata X C18 SPE column (Phenomenex, Torrance, CA, USA) and vacuum-dried. Control samples were labeled with TMT 126, 127, and 128, respectively. NaCl-treated samples were labeled with 129, 130, and 131, respectively. After incubation for 2 h at room temperature, samples were quenched, pooled, desalted, and dried by vacuum centrifugation.

For redox proteomics, another aliquot of protein samples was labeled with TMT reagents: 126, 127, and 128 for the control, and 129, 130, and 131 for the treated samples. After labeling, the samples were pooled, and the labeled proteins were digested, and enriched as described in the manufacturer’s instructions.

### 4.5. LC-MS/MS

Labeled peptides were then fractionated using HPLC (Thermo Betasil C18 column of 5 μm particles, 10 mm ID, 250 mm length). Peptides were dissolved in 0.1% formic acid (solvent A) and loaded onto a home-made reversed-phase analytical column. The gradient was comprised of an increase from 6% to 23% solvent B (0.1% formic acid in 98% acetonitrile) over 26 min, 23% to 35% in 8 min, and climbing to 80% in 3 min then holding at 80% for the last 3 min, at a constant flow rate of 400 nL/min on an EASY-nLC 1000 UPLC system. The peptides were subjected to an NSI source followed by tandem mass spectrometry (MS/MS) in Q Exactive Plus (Thermo) coupled online to the UPLC. The electrospray voltage was 2.0 kV. The *m*/*z* scan range was 350 to 1800 for the full scan in the Orbitrap at a resolution of 70,000. Peptides were then selected for MS/MS with the NCE of 28 and Orbitrap resolution of 17,500, fixed first mass of 100 *m*/*z*, and AGC target of 5E4.

### 4.6. Database Searching and Data Analysis

MS data were processed using Maxquant (v1.5.2.8). The database was *E. salsugineum* UniProt (28,349 sequences), and the decoy database was used to calculate the false positive rate (FDR). Other settings included: Trypsin/P for enzyme 2; for missing cleavages; 7 for the minimum peptide length; 5 for the maximum number of modifications; and 20 and 5 ppm parent ion and MS/MS tolerancefor the first search. Alkylation of cysteine was set as the fixed modification, and dynamic modifications included oxidation of methionine, acetylation of the N-terminal of protein, and deamidation (NQ). The quantitative method was set to TMT-6plex for global proteomics and redox proteomics. The FDR for protein identification and PSM identification was set to 1%. For global proteomics, the central median method was used for normalization. For redox proteomics, the relative abundance of cysteine-containing peptides was corrected based on the protein abundance.

GO was used for annotation (https://www.arabidopsis. org/tools/bulk/go/index.jsp (accessed on 4 July 2023)). Protein subcellular location was predicted using Wolfpsort (http://www.genscript.com/psort/wolf_psort.html (accessed on 9 July 2023)) and CELLO (http://cello.life.nctu.edu.tw/ (accessed on 9 July 2023)), and inconsistency was resolved by literature search. 

Log (base 2) transformed treatment/control ratios were used for hierarchical clustering analysis (http://bonsai.hgc.jp/~mdehoon/software/cluster/software.htm (accessed on 9 July 2023)). TBtools software (version 1.120) was used for data visualization. Statistical significance between the control and treatment was defined using fold change ≥ 2 and FDR < 0.01.

### 4.7. Chlorophylls

Chlorophyll fluorescence parameters were determined as described [38]. Briefly, 0.05 g of fresh leaves were cut into small pieces and transferred into a centrifugal tube. To each tube, 1 mL of 80% acetone was added. After incubation at 4 °C overnight, OD values were measured at 663, 645, and 440 nm, respectively. The following formulas were used for calculating the concentration of chlorophyll: chlorophyll a: C_a_ = 12.72·OD_663_ − 2.59·OD_645_; chlorophyll b: C_b_ = 22.88·OD_645_ − 4.67·OD_663_; carotenoid: CK = 4.7·OD_440_ − 0.27·(C_a_ + C_b_); total chlorophylls: C_T_ = C_a_ + C_b_.

### 4.8. NBT and DAB Staining of Leaves

Staining with nitrotetrazolium blue chloride (NBT) and 3,3-diaminobenzidine (DAB) were performed as described [39]. NBT was dissolved with 25 mM K-HEPES (pH 7.6) to 0.5 g/L. DAB was dissolved with 50 mM Tris-HAC (pH 5.0) to 1 g/L. Plants were immersed in freshly prepared NBT or DAB solutions and incubated overnight at 28 °C under dark. Samples were washed with 80% ethanol, boiled in a water bath for 10 min, and placed in anhydrous ethanol before image analysis.

### 4.9. Determination of the H_2_O_2_ and Malondialdehyde Contents

The levels of H_2_O_2_ in leaves were determined according to a previous report [39]. First, 0.1 g of leaves were ground in 1.0 mL of 10% TCA. After centrifuge at 15,000× rpm for 15 min, 4 °C, the supernatant was mixed with 0.5 mL of 10 mM PBS (pH 7.2) and 1 mL of 1 M KI. The mixture was incubated at 28 °C for 45 min under dark. The absorbance values of the supernatant were measured at 390 nm.

Malondialdehyde (MDA) was measured based on the formation of TBA-TCA conjugate [40]. In short, 0.1 g of leaves were ground in 1.9 mL of 10% TCA. To the supernatant, 1.5 mL of 0.6% thiobarbituric acid-trichloroacetic acid (TBA-TCA) was added. After reaction at 100 °C for 15 min, the absorbance values of the supernatant were measured.

### 4.10. Antioxidant Enzyme Activities

The enzyme activities were determined as in Bela et al. with modifications [9]. First, 0.1 g of fresh leaves were homogenized on ice in 1 mL of PBS. The homogenate was centrifuged for 20 min at 15,000× *g* at 4 °C, and the supernatant was used for enzyme activity assays. Superoxide dismutase (SOD) activity was determined spectrophotometrically by measuring the ability of the enzyme to inhibit the photochemical reduction of nitro blue tetrazolium (NBT) in the presence of riboflavin in light [41]. The reaction mixture contained 100 μL tissue extract, 50 mM PBS (PH = 7.8), 130 mM methionine, 0.75 mM NBT, 0.1 mM EDTA-Na2 and 0.02 mM Riboflavine in a total volume of 2.0 mL. The absorbance values of the supernatant were measured at 560 nm.

The catalase (CAT) activity was determined by the decomposition of H_2_O_2_ [42]. The reaction mixture contained 100 μL tissue extract, 30 mM H_2_O_2_ in 50 mM PBS (pH 7.8) in a total volume of 1.5 mL. One unit is the amount of decomposed H_2_O_2_ (μmol) in 1 min, the absorbance values of the supernatant were measured at 240 nm.

The ascorbate peroxidase (APX) activity was assayed according to Tari et al. [43]. The reaction mixture contained 20 mM H_2_O_2_, 100 μL tissue extract and 5 mM ascorbate in a potassium phosphate buffer (50 mM, pH 7.0) in a total volume of 2 mL. One unit was equal to nmol oxidized ascorbate in 30 s, and the absorbance values of the supernatant were measured at 290 nm.

The guaiacol peroxidase (POX) activity was determined by monitoring the increase in A470 during the oxidation of the guaiacol substrate according to Benyó et al. [41]. The reaction mixture contained 30 mM H_2_O_2_, 10 μL tissue extract and 20 mM substrate in a 50 mM PBS (pH 7.0) in a total volume of 1.5 mL. The amount of enzyme producing 1 μmol of oxidized guaiacol in 1 min was defined as one unit.

The peroxidase (POD) activity was determined by monitoring the increase in A470 during the oxidation of the guaiacol substrate [40]. The reaction mixture contained 40 mM H_2_O_2_, 100 μL tissue extract and 0.1 M substrate in 50 mM PBS (pH 7.0) in a total volume of 3.0 mL. The amount of enzyme producing 1 μmol of oxidized guaiacol in 5 min was defined as one unit.

The glutathione transferase (GST) enzyme activity, GR activity, total glutathione and oxidized glutathione (GSSG) concentrations were measured according to a reagent kit (sheng gong; NO. D799613; NO. D700615; NO. D799261; NO. D799612).

## 5. Conclusions

In this study, 107 redox proteins were identified from the tandem mass tag (TMT)-based proteomics. After a comprehensive analysis of the protein expression and GO analysis, we determined that GPX had the greatest potential in regulating salt tolerance.

## Figures and Tables

**Figure 1 ijms-24-14518-f001:**
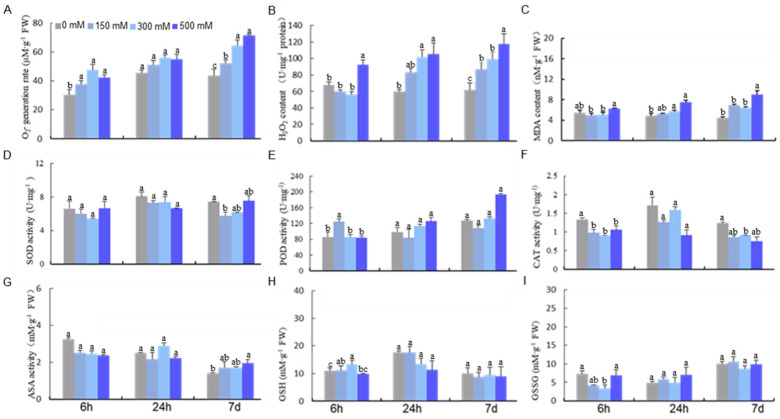
Effect of NaCl treatment on the activities of antioxidant-related enzymes and contents of reductants in *Eutrema salsugineum*. (**A**) O_2_^−^ generation rate; (**B**) H_2_O_2_ content; (**C**) malondialdehyde content (MDA); (**D**) peroxidase (POD); (**E**) superoxide dismutase (SOD); (**F**) catalase (CAT); (**G**) reduced ascorbate (AsA); (**H**) reduced glutathione (GSH); (**I**) glutathione disulfide oxidized glutathione, experiments were repeated at least three times with similar results. Data are presented as mean ± SD (*n* = 3). Different letters denote significant differences at *p* < 0.05 (one-way ANOVA, Tukey’s test).

**Figure 2 ijms-24-14518-f002:**
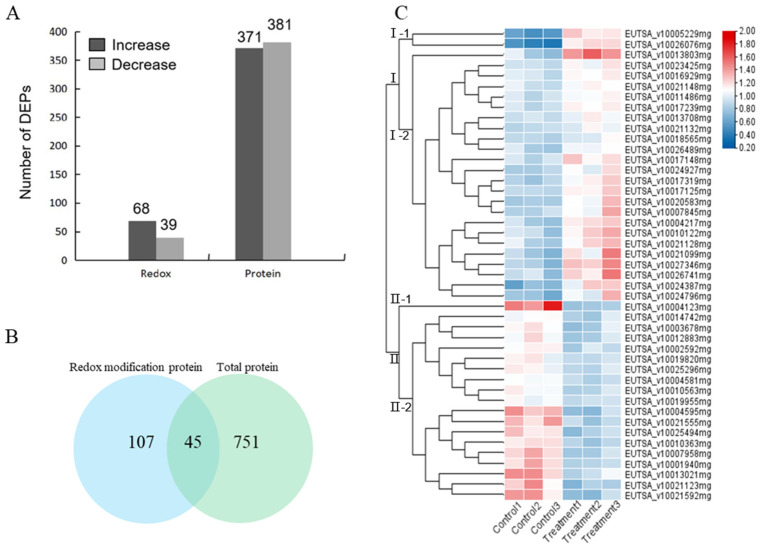
Distribution of differentially modified proteins in different comparison groups. (**A**) Numbers of differentially expressed proteins. (**B**) Overlap of differentially oxidized proteins identified and total protein. (**C**) Dendrogram of 45 redox-regulated proteins obtained by hierarchical clustering analysis. Two main clusters (I and II) and subclusters of I and II (I-1, I-2, II-1, and II-2) are showed on the left side. Protein name abbreviations are listed on the right side. The increased and decreased proteins are represented in red or blue, respectively. The color intensity increases with increasing abundance differences, as shown in the scale bar. Detailed information for protein names can be found in Appendix A.

**Figure 3 ijms-24-14518-f003:**
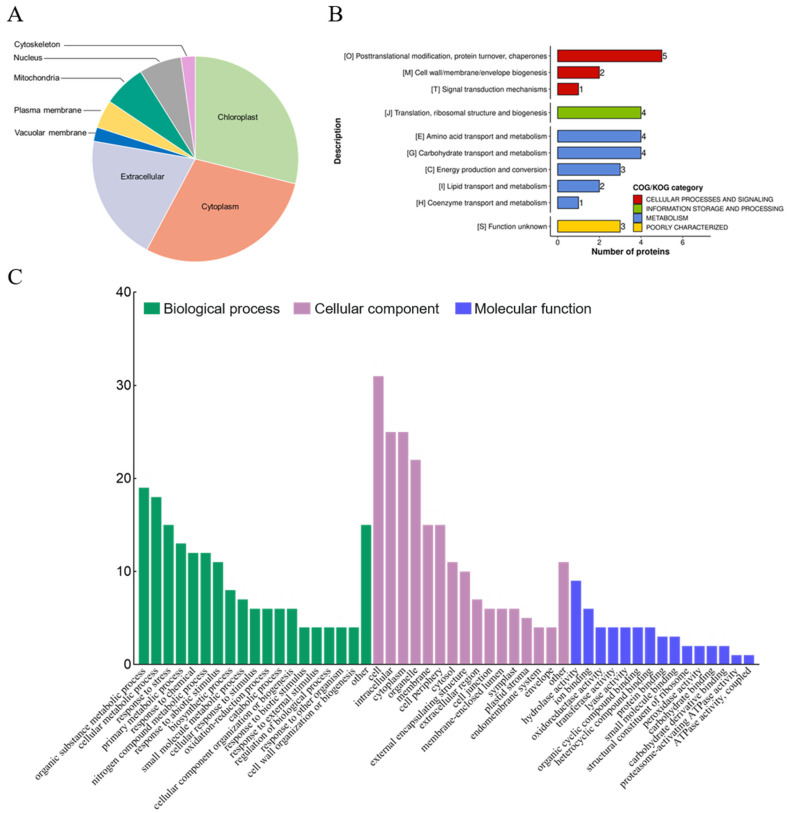
Identification of subcellular localization and functional characterization of redox proteins in *Eutrema salsugineum* leaves. (**A**) Pie chart of subcellular localization of redox proteins. (**B**,**C**) Clusters of orthologous groups (COG) and Gene Ontology (GO) and enrichment analysis of redox proteins for the biological process, molecular function, and cellular component categories.

**Figure 4 ijms-24-14518-f004:**
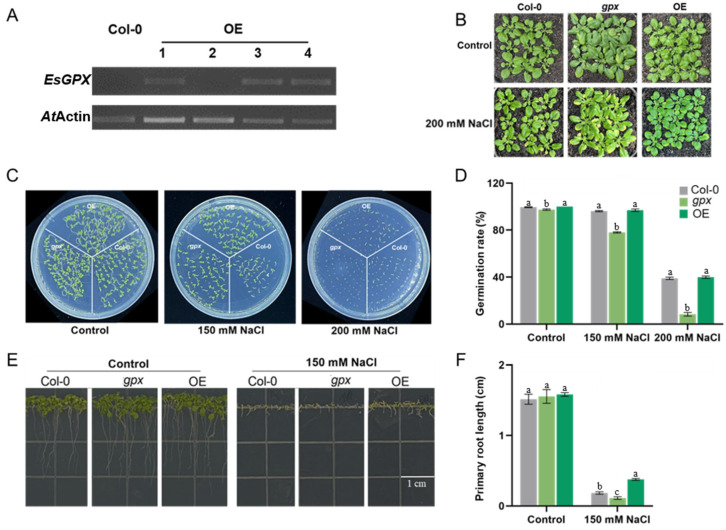
Seed germination and root elongation of *Arabidopsis* seedlings overexpressing *EsGPX* under salt stress. (**A**) Gene expression of *EsGPX* in the transgenic and WT plants. (**B**) Phenotypes of five-week-old plants grown in soil (four weeks under normal condition, followed by one-week NaCl treatment). (**C**,**D**) Seed germination under NaCl treatment. (**E**,**F**) Root elongation under NaCl treatment. Scale bars are 1 cm for roots. Different letters denote significant differences at *p* < 0.05 (one-way ANOVA, Tukey’s test).

**Figure 5 ijms-24-14518-f005:**
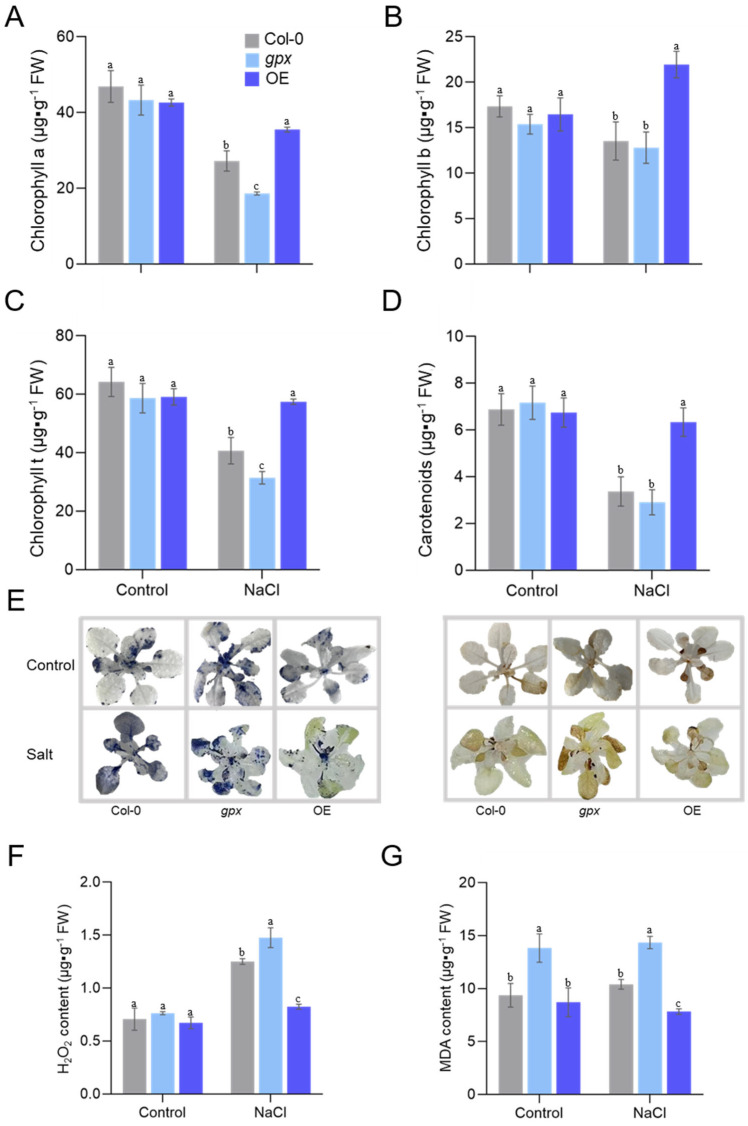
Effects of salt stress on H_2_O_2_ and MDA contents in leaves of *Arabidopsis* with overexpression of *EsGPX* genes and wild−type. (**A**–**D**) The contents of chlorophyll a, chlorophyll b, total chlorophyll and carotenoids in leaves of wild-type and overexpressed *EsGPX* under normal and salt stress. (**E**) NBT and DAB staining of wild-type and overexpressed *EsGPX*. (**F**,**G**) The contents of H_2_O_2_ and MDA in *Arabidopsis* leaves of wild-type and overexpressed *EsGPX* gene. Data are presented as mean ± SD (*n* = 3). Different letters denote significant differences at *p* < 0.05 (one-way ANOVA, Tukey’s test).

**Figure 6 ijms-24-14518-f006:**
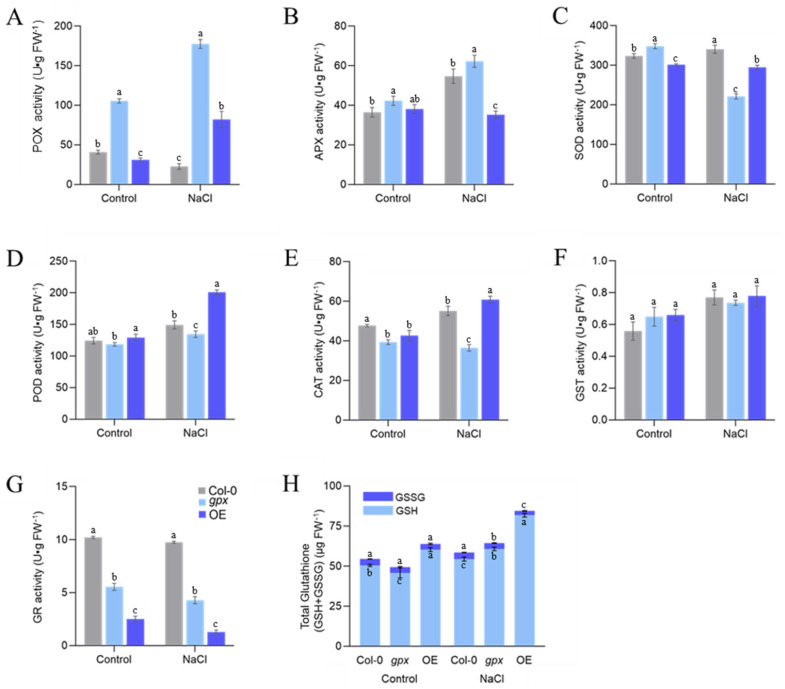
Effect of one-week 200 mM NaCl treatment on the guaiacol peroxidase (**A**), ascorbate peroxidase (**B**), superoxide dismutase (**C**), peroxidase, (**D**) catalase, (**E**) glutathione transferase, (**F**) glutathione reductase, (**G**) glutathione, and (**H**) activities in roots and shoots, respectively, of 5−week-old *Arabidopsis thaliana* wild type (Col−0), *gpx* mutant and OE−*EsGPX* plants. Data are presented as mean ± SD (*n* = 3). Different letters denote significant differences at *p* < 0.05 (one−way ANOVA, Tukey’s test).

## Data Availability

All the data that support the findings of this study are available in the paper and its Appendix A published online.

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
