# Peer review of "Redox Regulation of Salt Tolerance in Eutrema salsugineum by Proteomics"

_ijms, 2023, doi:10.3390/ijms241914518_

Round 1

Reviewer 1 Report

The study was focused on redox regulation of NaCl response in Eutrema salsugineum revealed by redox proteomics and ectopic expression of EsGPX. The Authors revealed that salt stress led to increased oxidative modification levels of 159 cysteine sites in 107 proteins, which play roles in carbohydrate and energy metabolism, transport, ROS homeostasis, cellular structure modulation, and folding and assembly. These lists of unknown redox reactive proteins in salt mustard lay the foundation for future research to understand the molecular mechanism of plant salt response.

The paper is quite interesting, however, I recommend few revisions:

-        Table 1 should be removed into the Supplementary File,

-        Regarding statistical analyses: In most cases, factorial ANOVA test with subsequent post-hoc test (e.g. Tukey’s test) should be re-calculated,

-        Discussion and Conclusions parts should be revised into a more in-depth interpretation of the obtained results,

-        Extensive editing of English language is required.

Extensive editing of English language is required.

Reviewer 2 Report

The manuscript by Jiawen Li et al. presents two conected experiments aimed at redox regulatory pathways underyling response to salt stress. 

Positives:

The authors measured enzyme activity, profiled proteome response, and constructed a glutathione peroxide overexpressor line with an improved salt tolerance.

Negatives:

The manuscript is far from the first study of this type (e.g., 10.1016/j.bbabio.2021.148482; 10.1111/ppl.12248; 10.3389/fpls.2022.909527;10.1186/s40529-022-00337-w). Its style and language require significant polishing, experiment design and descriptions are incomplete, the integration of mutant/overexpressor in the study is not clear. It is possible that the manuscript could be modfied and supplemented to address my issues, but I don't consider that to be only a major revision.  

Major issues:

1/ The title of the manuscript and its abstract promisse to deliver an insight into the redox regulatory mechanisms via REDOX proteomics. However, that part is severly limited. The method description is incomplete/missing, statistical evaluation is not clear, and the selected 1.2-fold change is below generally accepted threshold for proteomics analyses.

The identification of redox regulations requires quantitation of the corresponding proteins. That part is very confusing, and the method must be clearly described and the corresponding statistics included.

Table 1 - Is the TMT FC based on estimated protein abundance or the estimated total abundance of C-containing peptide (as it should be)?

The listed standard deviations are extremly high (e.g., 0.163±1.735), and it is not very likely that the presented differences are statistically significant for all listed values. That issue must be addressed, and only statistically significant differences should be presented. 

2/ The experiment with EsGPX is interesting, but how does it fit the redox regulation of proteome? Table 1 indicates that the abundance of this enzyme is drastically lower under salt stress (0.051), indicating that it is likely a negative regulator of salt stress tolerance. That would not justify its selection for overexpression in A. thaliana. The general impact of ROS disbalance via peroxidase overexpression on salt tolerance is well known and not novel (e.g., 10.3390/ijms20153745). However, it is not clear how the overexpression of GPX impacts its oxidation and how does the oxidation impact GPX activity. That would be interesting topic to address, matching the topic of REDOX regulation, but none of that is found in the manuscript.

That said, the mutant description per se seems to be interesting, but it should be supplemented with more data. For instance, a single mutant line is not sufficient, unless the authors can provide evidence that the construct insertion did not impact a different gene/its regulation.

I also note that phenotype of OE line at 150 mM in panel C does not seem to match that of panel E. 

Minor issues

- Protein abundance should not be missplaced with "expression".

- The proteomics data should be deposited into a public repository.

- The employed pair-wise comparisons are not sufficient for plots comparing more than two genotypes/treatments. A suitable statistical test must be employed.

- The enzyme activity normalization should be done on dry weight, given the NaCl impact on water balance and FW. 

- Figure legends should include information about reproducibility (number of biological replicates).

The manuscript requires extensive language editing.

Round 2

Reviewer 2 Report

The authors have addressed some of my major issues by removing data to supplements (Table 1). However, the evaluation of differences and statistics in Table S1 should be improved. 

Issues not fully addressed in the revision:

That said, the mutant description per se seems to be interesting, but it should be supplemented with more data. For instance, a single mutant line is not sufficient, unless the authors can provide evidence that the construct insertion did not impact a different gene/its regulation.

My issue was with both mutant lines - the overexpressor and the loss-of-function one. 

The following minor issues were not reflected in the revision:

- Protein abundance should not be missplaced with "expression".

- The proteomics data should be deposited into a public repository.

Language nees improvment, including the revised title. 
